# GAL4-based functional screen of neuropeptides in *Drosophila* reproduction

**Madhumala K. Sadanandappa**[¤a]*, **Caliope Marin**[¤b], **Shinae Park**, **Shivaprasad H. Sathyanarayana**[¤a], **Giovanni Bosco**[*]

Department of Molecular and Systems Biology, Geisel School of Medicine at Dartmouth, Hanover, New Hampshire, United States of America

¤a Current address: Department of Pathology and Laboratory Medicine, Dartmouth-Hitchcock Medical Center, Lebanon, New Hampshire, United States of America
¤b Current address: Department of Neurobiology, School of Biological Sciences, University of California San Diego, La Jolla, California, United States of America
* Madhumala.K.Sadanandappa@hitchcock.org (MKS); Giovanni.Bosco@dartmouth.edu (GB)

## Abstract

Neuropeptides are evolutionarily conserved signaling molecules that regulate diverse behavioral and physiological processes, including reproduction. Although, several neuropeptides have established roles in reproductive regulation, the reproductive functions of many neuropeptides in *Drosophila melanogaster* remain poorly characterized. Here, we performed a targeted neurogenetic screening to systematically assess the contribution of 25 neuropeptides to reproductive output. Using neuropeptide-specific GAL4 drivers and synaptic silencing with tetanus toxin, we quantified the egg-laying as an integrated functional readout of reproduction. Disruption of 14 neuropeptides altered egg-laying, including eight neuropeptides not previously described to play roles in reproductive regulation. While some of these effects are likely indirect and may reflect contributions from both female and male flies or systematic physiological signaling, these results reveal broad involvement of neuropeptidergic pathways in reproductive function. Collectively, this study establishes a functional screening framework, identifies new reproductive neuropeptides, and provides a curated resource to guide future mechanistic studies of neuropeptide-mediated brain-gonad communication.

## Introduction

Neuropeptides are evolutionarily conserved signaling molecules that coordinate diverse behavioral and physiological processes, including feeding, metabolism, circadian rhythms, stress responses, and reproduction [1–5]. Unlike classical neurotransmitters, neuropeptides often act over longer spatial and temporal scales, enabling the integration of complex systemic physiological signals across tissues. Neuropeptides are expressed not only in the nervous system but also in various peripheral

**Data availability statement:** All raw datasets generated in this study are provided in the Supplemental information files. Further enquires can be directed to the corresponding authors: Madhumala.K.Sadanandappa@hitchcock.org (MKS) and Giovanni.Bosco@dartmouth.edu (GB).

**Funding:** This work was supported by the Human Frontier Science Program [LT000933/2017 to MKS] and the National Institute of Health [Pioneer grant 1DP1MH110234 to GB]. The funders had no role in study design, data collection and analysis, decision to publish, or preparation of the manuscript.

**Competing interests:** The authors have declared that no competing interests exist.

and internal organs, contributing to autonomic and homeostatic functions [1,6]. In *Drosophila melanogaster*, over 50 neuropeptides and their cognate receptors have been identified, many structurally and functionally conserved with vertebrate counterparts [5–12]. This high degree of conservation, combined with versatile genetic tools, makes *Drosophila* an excellent model for investigating neuropeptide-mediated regulation of reproduction and brain-gonad communication.

Reproduction is tightly coordinated by internal physiological cues and environmental factors across species [1,3]. In mammals, the hypothalamic-pituitary-gonadal axis centrally regulates reproduction, wherein hypothalamic gonadotrophin-releasing hormone (GnRH) stimulates the release of follicle-stimulating hormone and luteinizing hormone from the anterior pituitary, which in turn control gametogenesis and hormone production [13,14]. GnRH secretion is further modulated by neuropeptides such as kisspeptin and neuropeptide Y (NPY), as well as metabolic hormones including leptin and insulin, linking reproductive function to energy balance and environmental conditions [15]. Similarly, in *Drosophila*, neuropeptides act directly on reproductive tissues and germs cells, and indirectly by regulating reproductive behaviors, nutritional status, and systemic physiology. While classical regulators such as ecdysteroids, juvenile hormone (JH), sex peptide (SP), neuropeptide F (NPF), and *Drosophila* insulin-like peptides (DILPs) have been well characterized, the roles of many other neuropeptides in reproduction remain poorly described [6,16–18].

To systematically investigate neuropeptide contributions to reproduction, we performed a targeted neurogenetic screen of 25 neuropeptide genes in *Drosophila* (Fig 1). These were selected in two categories: (i) neuropeptides described or known to have a role in reproduction, to validate the sensitivity and robustness of our functional assay, and (ii) neuropeptides not previously described in reproduction, enabling discovery of novel regulators. This design allowed both confirmatory testing of known reproductive neuropeptides and unbiased identification of novel candidates.

We used neuropeptide-specific GAL4 drivers combined with synaptic silencing via tetanus toxin and quantified the number of eggs laid as a functional readout of reproductive output. This approach minimized off-target effects commonly associated with RNAi-mediated perturbations while allowing assessment of both direct actions on reproductive tissues and indirect effects mediated through systemic physiology and behavior [6,9,19–21]. Overall, our study provides a functional framework for linking neuropeptidergic signaling to reproductive function, validates known regulators, and identifies novel candidate neuropeptides for mechanistic studies of brain-gonad communication and conserved neuropeptide functions across species.

## Materials and methods

### *Drosophila* stocks and fly husbandry

Unless otherwise stated, all fly lines, including crosses, were maintained on standard cornmeal medium composed of cornmeal, molasses, agar, and yeast [22], at 25 ℃ under 12:12 hours light-dark (LD) cycle-controlled incubators. Except for *Tk-gut-GAL4* (Irene Miguel-Aliaga, Imperial College London, UK) and *UAS-mCD8::GFP* (Mani Ramaswami, Trinity College Dublin, Ireland), all other listed fly lines were

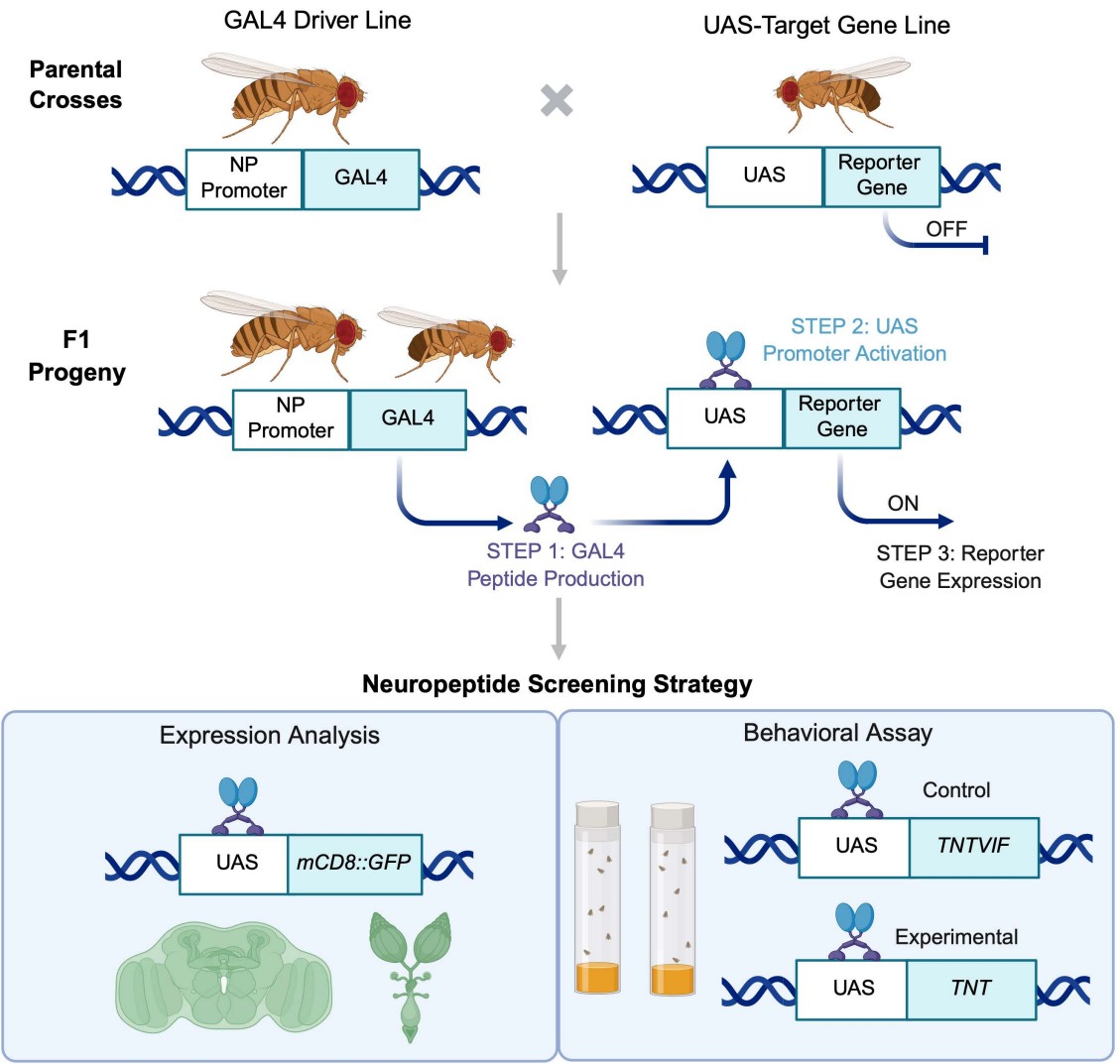

**Fig 1. Workflow for the *Drosophila* neuropeptide screen.** Schematic representation of the experimental workflow used for functional screening of neuropeptides using the GAL4-UAS system. Unmated *NP-GAL4* females were crossed with UAS-transgene males, and F1 female progeny were analyzed for neuropeptide expression in the brain and ovaries using *UAS-mCD8::GFP* reporter. To examine neuropeptide function in reproduction, an egg-laying assay was performed using genotype- and age-matched flies expressing either an inactive (*UAS-TNTVIF*) or active (*UAS-TNT*) form of tetanus toxin. Schematic created with BioRender.com.

obtained from the Bloomington *Drosophila* Stock Center (BDSC; https://bdsc.indiana.edu). For stock information, refer to Table 1. Additional genetic information is available on FlyBase (http://flybase.org).

## Immunostaining

Brains and ovaries were dissected from 6-days-old, mated females and immunolabeled following a previously described protocol [23]. Primary antibodies included chicken anti-GFP (1:1000, #ab13970, Abcam, MA, USA) and mouse anti-Bruchpilot (nc82, 1:20, Erich Buchner, University of Würzburg, Germany). Alexa fluor-conjugated secondary antibodies (1:400) were used for detection. Ovaries were stained with phalloidin and 4',6-diamidino-2-phenylindole (DAPI).

**Table 1.** *Drosophila* stocks used in this study.

| | Neuropeptide (Acronym) | Genotype | Identifier | Chromosome insertion |
|---|---|---|---|---|
| 1 | Adipokinetic Hormone (AKH) | *y[1] w[*]; P{w[+mC]=Akh-gal4.L}2/CyO, y[+]*<br>*y[1] w[*]; P{w[+mC]=Akh-gal4.L}3* | RRID:BDSC_25683<br>RRID:BDSC_25684 | 2<br>3 |
| 2 | Allatostatin A (AstA) | *w[1118]; P{w[+mC]=AstA-GAL4.2.1}3M/TM6B, Tb [1]*<br>*w[1118]; P{w[+mC]=AstA-GAL4.2.1}5*<br>*w[1118]; wg[Sp-1]/CyO; P{w[+mC]=AstA-GAL4.2.74}4* | RRID:BDSC_51978<br>RRID:BDSC_51979<br>RRID:BDSC_80160 | 3<br>2<br>3 |
| 3 | Allatostatin B (AstB/MIP) | *w[1118]; P{w[+mC]=Mip-GAL4.TH}1M/TM6B, Tb [1]*<br>*w[1118]; P{w[+mC]=Mip-GAL4.TH}2M* | RRID:BDSC_51983<br>RRID:BDSC_51984 | 3<br>2 |
| 4 | Allatostatin C (AstC) | *w[1118]; P{w[+mC]=AstC-GAL4.TH}1M/TM6B, Tb [1]* | RRID:BDSC_52017 | 3 |
| 5 | Bursicon (Burs) | *w[1118]; P{w[+mC]=Burs-GAL4.TH}4M*<br>*w[*]; P{w[+mC]=Burs-GAL4.P}P12* | RRID:BDSC_51980<br>RRID:BDSC_40972 | 2<br>2 |
| 6 | Partner of bursicon (pBurs) | *w[1118]; PBac{w[+mC]=IT.GAL4}1139-G4* | RRID:BDSC_65470 | 2 |
| 7 | Capability (CAPA) | *w[1118]; P{w[+mC]=Capa-GAL4.TH}4F*<br>*w[1118]; P{w[+mC]=Capa-GAL4.TH}5F* | RRID:BDSC_51969<br>RRID:BDSC_51970 | 2<br>X |
| 8 | CCHamide-1 (CCHa-1) | *w[1118]; Mi{GFP[E.3xP3]=ET1}CCHa1[MB11962]* | RRID:BDSC_29266 | 3 |
| 9 | Crustacean cardioactive peptide (CCAP) | *y[1] w[*]; P{w[+mC]=CCAP-GAL4.P}16/CyO*<br>*y[1] w[*]; Bl [1]/CyO, y[+]; P{w[+mC]=CCAP-GAL4.P}9* | RRID:BDSC_25685<br>RRID:BDSC_25686 | 2<br>3 |
| 10 | Corazonin (CRZ) | *w[1118]; P{w[+mC]=Crz-GAL4.391}3M*<br>*w[1118]; P{w[+mC]=Crz-GAL4.391}4M* | RRID:BDSC_51976<br>RRID:BDSC_51977 | 2<br>3 |
| 11 | dFMRFamide (dFMRFa) | *w[1118]; P{w[+mC]=FMRFa-GAL4.TH}1M*<br>*w[1118]; P{w[+mC]=FMRFa-GAL4.S}FG5* | RRID:BDSC_51990<br>RRID:BDSC_56837 | 3<br>2 |
| 12 | Diuretic hormone 31 (DH31) | *w[1118]; P{w[+mC]=Dh31-GAL4.TH}2M*<br>*w[1118]; P{w[+mC]=Dh31-GAL4.TH}5F* | RRID:BDSC_51988<br>RRID:BDSC_51989 | 3<br>X |
| 13 | Diuretic hormone 44 (DH44) | *w[1118]; P{w[+mC]=Dh44-GAL4.TH}2M* | RRID:BDSC_51987 | 3 |
| 14 | Drosulfakinin (DSK) | *w[1118]; P{w[+mC]=Dsk-GAL4.TH}3M* | RRID:BDSC_51981 | 3 |
| 15 | Ecdysis triggering hormone (ETH) | *w[1118]; P{w[+mC]=ETH-GAL4.TH}1M* | RRID:BDSC_51982 | 2 |
| 16 | Eclosion hormone (EH) | *P{w[+mC]=GAL4-Eh.2.4}C21* | RRID:BDSC_6301 | 2 |
| 17 | Hugin (hug-PK) | *w[*]; P{w[+mC]=Hug-GAL4.S3}3* | RRID:BDSC_58769 | 3 |
| 18 | Leucokinin (LK) | *w[1118]; P{w[+mC]=Lk-GAL4.TH}1*<br>*w[1118]; P{w[+mC]=Lk-GAL4.TH}2M* | RRID:BDSC_51992<br>RRID:BDSC_51993 | X<br>2 |
| 19 | Myosuppressin (MS) | *w[1118]; P{w[+mC]=Ms-GAL4.TH}1M/TM6B, Tb [1]*<br>*w[1118]; P{w[+mC]=Ms-GAL4.TH}6Ma* | RRID:BDSC_51985<br>RRID:BDSC_51986 | 3<br>2 |
| 20 | Pigment-dispersing factor (PDF) | *P{w[+mC]=Pdf-GAL4.P2.4}X, y[1] w[*]* | RRID:BDSC_6899 | X |
| 21 | Proctolin (Proc) | *w[1118]; P{w[+mC]=Proc-GAL4.TH}2M/TM6B, Tb [1]*<br>*w[1118]; P{w[+mC]=Proc-GAL4.TH}6M* | RRID:BDSC_51971<br>RRID:BDSC_51972 | 3<br>2 |
| 22 | RYamide (RYa) | *w[1118]; PBac{w[+mC]=IT.GAL4}0922-G4* | RRID:BDSC_63899 | 2 |
| 23 | short Neuropeptide F (sNPF) | *P{w[+mC]=sNPF-GAL4.TH}2, w[1118]* | RRID:BDSC_51991 | X |
| 24 | SIFamide (SIFa) | *w[1118]; SIFa[1]*<br>*w[1118]; SIFa[2]*<br>*w[1118]; SIFa[3]* | RRID:BDSC_80696<br>RRID:BDSC_80697<br>RRID:BDSC_80698 | 2<br>2<br>2 |
| 25 | Tachykinin (TK) | *Tk-gut-GAL4* | N/A | 2 |
| | **Other stocks** | | | |
| 26 | *w[*];P{w[+mC]=UAS-TeTxLC.(-)V}A2* | | RRID:BDSC_28840 | 2 |
| 27 | *w[*];P{w[+mC]=UAS-TeTxLC.tnt}G2* | | RRID:BDSC_28838 | 2 |
| 28 | *UAS-mCD8::GFP* | | N/A | 2 |

Samples were mounted in Vectashield (H-1000, Vector Laboratories, CA, USA) and imaged using a Nikon A1R confocal microscope or a Nikon Eclipse E800 microscope. Acquired fluorescent images were processed with Nikon NIS-Elements and Adobe Photoshop. For each genotype and neuropeptide, 10 brains and 10 ovaries were examined for GAL4 driven *UAS-mCD8::GFP* expression.

### Behavioral assay

Freshly emerged F1 progeny (0–12 hrs post-eclosion) from control (*NP-GAL4 > UAS-TNTVIF*) and experimental (*NP-GAL4 > UAS-TNT*) crosses were collected and transferred to separate bottles containing fresh fly media (~50 flies per bottle). In each experiment, *NP-GAL4* refers to the neuropeptide-specific GAL4 insertion line being tested (Table 1). Flies were kept for six days at 25 ºC under a 12:12 hours LD cycle.

For the behavioral assay, 6-days-old flies were anesthetized with carbon dioxide, and groups of five females and two males of the same genotype–unless otherwise indicated–were placed in vials with fresh media. After 24-hours of egg-laying under controlled incubator conditions, adult flies were removed, and eggs were counted using a ZEISS Stemi 2000 stereomicroscope [24]. All vials were coded, and the experimenter was blinded to the genotype tested.

### Statistical analysis

Data analysis and visualization were performed using Microsoft® Excel (version 16.90) and GraphPad Prism (version 10.2.3). Mean values were derived from two to three independent experiments. Statistical significance was assessed using an unpaired *t*-test with Welch's correction. The following significance thresholds were applied: $p$-value $< 0.05$ (*), $p < 0.001$ (**), $p < 0.0001$ (***), $p < 0.00001$ (****), and $p > 0.05$ was considered non-significant (ns).

## Results and discussion

### Strategy for neuropeptide screening

This neurogenetic screen was designed as an unbiased functional survey to assess the contribution of *D. melanogaster* neuropeptides to reproductive output using a genetic and behavioral framework. We selected 25 neuropeptides known to regulate diverse behavioral and physiological processes, including foraging, aggression, sensory processing, circadian rhythms, sleep, growth and development, ecdysis, lifespan, metabolism, diapause, stem cell activation and homeostasis, reproduction, stress responses, learning and memory [5–12]. An overview of these neuropeptides, their receptors, and previously reported functions is provided in Table 2.

The selected 25 neuropeptides were grouped into two categories. The first included neuropeptides with known roles in reproduction to validate the robustness and sensitivity of the assay. The second included neuropeptides for which reproductive functions had not been previously described, allowing the identification of potential novel regulators of reproduction in *Drosophila*. Well-characterized reproductive neuropeptides, such as DILPs, SP, and NPF, were excluded from the current screen to focus on additional candidate regulators [6,9,17].

To examine neuropeptidergic contribution to reproductive output, we employed GAL4/UAS-based neurogenetic approach [190]. Neuropeptide-specific GAL4 driver lines (*NP-GAL4*) were first validated by assessing the expression of a membrane-bound fluorescent reporter (*UAS-mCD8::GFP*) in the brains and ovaries of adult female flies (Fig 1, Tables 1 and 3, and S1 File). This expression analysis was performed to primarily to confirm driver activity and provide anatomical context for the screening results, rather than to establish functional causality. Following validation, reproductive output was quantified using a standard 24-hour egg-laying assay.

Neuropeptide signaling was disrupted by expressing the active form of tetanus toxin (*UAS-TNT*) under *NP-GAL4* control, thereby blocking synaptic vesicle release through enzymatic cleavage of synaptobrevin–a core component of the SNARE (soluble N-ethylmaleimide-sensitive factor attachment protein receptor) [19,20]. Age- and genotype-matched flies

**Table 2.** *Drosophila* neuropeptides, their receptors, and previously reported functions**.

| | Neuropeptide (Acronym) | Annotation | Receptor(s) | Receptor annotation | Reported functions** | References |
|---|---|---|---|---|---|---|
| 1 | Adipokinetic Hormone (AKH) | CG1171 | AkhR | CG11325 | Regulates nutritional and oxidative stress responses, carbohydrates and lipids homeostasis, locomotor activity, diapause, and lifespan | [25–39] |
| 2 | Allatostatin A (AstA) | CG13633 | AstA-R1 AstA-R2 | CG2872 CG10001 | Involved in food search and feeding behaviors, sleep, appetitive learning, juvenile growth and maturation | [40–46] |
| 3 | Allatostatin B (AstB/MIP) | CG6456 | SPR | CG16752 CG30106# CG14484# | Modulates chemosensory processing, feeding behaviors, circadian clock, sleep, and female mating receptivity | [47–53] |
| 4 | Allatostatin C (AstC) | CG14919 | AstC-R1 AstC-R2 | CG7285 CG13702 | Food intake, metabolic homeostasis, nociception, circadian rhythm for locomotor activity and oogenesis, diapause, and reproduction | [54–59] |
| 5 | Bursicon (Burs/Burs alpha) | CG13419 | rickets (rk) | CG8930 | Functions in cuticle tanning and sclerotization, wing expansion, energy homeostasis, and sleep plasticity | [60–64] |
| 6 | Partner of bursicon (pBurs/Burs beta) | CG15284 | rickets (rk) | CG8930 | Functions in cuticle tanning and sclerotization and wing expansion | [61,62] |
| 7 | Capability (CAPA) | CG15520 | CapaR PK1-R | CG14575 CG9918 | CAPA-PVK1 and 2 acts as diuretic hormone on Malpighian tubules, involved in osmoregulation, myostimulation, and stress response | [65–69] |
| 8 | CCHamide-1 (CCHa-1) | CG14358 | CCHa1-R | CG30106 | Regulates circadian activity and sleep, sensory perception, and olfactory behaviors | [70–74] |
| 9 | Crustacean cardioactive peptide (CCAP) | CG4910 | CCAP-R | CG6111 | Ecdysis, heartbeat regulation, feeding behaviors and metabolism | [75–78] |
| 10 | Corazonin (CRZ) | CG3302 | CrzR | CG10698 | Modulates food search and feeding behaviors, ethanol seduction and metabolism, stress responses, sexually dimorphic behaviors, sperm transfer and copulation | [79–86] |
| 11 | dFMRFamide (dFMRFa) | CG2346 | dFMRFaR | CG2114 | Controls ecdysis, sleep, myomodulation, body fat, and flight behaviors | [87–92] |
| 12 | Diuretic hormone 31 (DH31) | CG13094 | DH31-R | CG32843 CG4395 | Diuretic peptide required for larval peristalsis, feeding-courtship behavioral switch, reproductive dormancy, circadian control of locomotor activity, temperature preference, sleep, learning and memory | [93–102] |
| 13 | Diuretic hormone 44 (DH44) | CG8348 | DH44-R1 DH44-R2 | CG8422 CG12370 | Regulates locomotor activity, diuresis, nutrient-sensing, circadian control of activity-rest rhythms, starvation tolerance, sperm ejection and storage | [103–116] |
| 14 | Drosulfakinin (DSK) | CG18090 | CCKLR-17D1 CCKLR-17D3 | CG42301 CG32540 | Controls feeding, locomotor activity, nociception, aggression, social and reproductive behaviors | [117–126] |
| 15 | Ecdysis-triggering hormone (ETH) | CG18105 | ETHR | CG5911 | Molting exocuticle (ecdysis), reproduction, courtship behaviors and memory | [127–132] |
| 16 | Eclosion hormone (EH) | CG5400 | Guanylyl cyclase* | CG10738 | Development and ecdysis behaviors | [133–135] |
| 17 | Hugin (hug-PK) | CG6371 | PK2-R1 PK2-R2 | CG8784 CG8795 | Taste, food search and intake, evasion behavior, circadian rhythm, and heart rate | [111,136–141] |

*(Continued)*

**Table 2.** (Continued)

| | Neuropeptide (Acronym) | Annotation | Receptor(s) | Receptor annotation | Reported functions** | References |
|---|---|---|---|---|---|---|
| 18 | Leucokinin (LK) | CG13480 | LKr | CG10626 | Modulates by chemosensory responses, feeding behaviors, diuresis, circadian activity, airway clearance, stress tolerance, and escape behaviors | [113,142–152] |
| 19 | Myosuppressin (MS) | CG6440 | MsR1 MsR2 | CG8985 CG43745 | Circadian activity, sleep, muscle contraction, food search and intake | [153–155] |
| 20 | Pigment-dispersing factor (PDF) | CG6496 | Pdfr | CG13758 | Synchronization and regulation of clock neurons activity, circadian rhythms, locomotion, diuresis, mating and copulation | [156–161] |
| 21 | Proctolin (Proc) | CG7105 | Proc-R | CG6986 | Controls heart rate, muscle contraction, and larval locomotion | [162–164] |
| 22 | RYamide (RYa) | CG40733 | RYa-R | CG5811 | Suppress feeding behaviors | [165,166] |
| 23 | short Neuropeptide F (sNPF) | CG13968 | sNPF-R | CG7395 | Regulates cell and organism growth, carbohydrate metabolism, feeding behaviors, nociception, olfactory processing, circadian rhythm, locomotion, sleep, lifespan, learning and memory | [158,160,167–176] |
| 24 | SIFamide (SIFa) | CG33527 | SIFaR | CG10823 | Modulates appetite, feeding rhythm, sleep, and sexual behaviors, including female receptivity to male courtship | [177–182] |
| 25 | Tachykinin (TK) | CG14734 | TkR86C TkR99D | CG6515 CG7887 | Lipid metabolism, aggression behavior, nociception, olfactory processing, locomotion, and food search | [183–189] |

**Note:** *Indicates non-GPCR receptors; #putative receptors; **This table presents a representative selection of functions and references for *Drosophila* neuropeptides but is not an exhaustive list.

expressing an inactive tetanus toxin (*UAS-TNTVIF*) under the same *NP-GAL4* drivers served as controls (Tables 1 and 3). This strategy enabled a systematic functional evaulation of neuropeptidergic signaling in reproductive output (Figs 1 and 2).

## Outcomes of neuropeptide screening

Reproductive output in *Drosophila* depends on coordinated regulation of behavior and physiology, including courtship, mating, gametogenesis, and post-mating responses, which collectively promote reproductive success [6]. Female oogenesis is a complex, multistep process regulated by hormonal, genetic, and neuronal inputs that collectively govern follicle maturation, oocyte polarity, and egg release [17,191].

Given this complexity, changes observed in behavioral assay may reflect either direct effects of female reproductive physiology or indirect influences mediated through neural circuits, endocrine pathways, or male-derived contributions. Improtantly, because both male and female flies in the experimental groups carried *NP-GAL4 > UAS-TNT* transgenes, the observed phenotypes could arise from neuropeptide perturbation in females, males, or both. Accordingly, we interpret the results as identifying neuropeptides with potential roles in reproduction rather than assigning sex-specific or mechanistic functions.

Based on phenotypic outcomes from neuronal silencing using the GAL4/UAS system, neuropeptides were classified into four categories:

a. **Reduced egg-laying:** Disruption of nine neuropeptides−allatostatin B (AstB/MIP), bursicon (Burs), capability (CAPA), corazonin (CRZ), diuretic hormone 44 (DH44), drosulfakinin (DSK), *Drosophila* FMRFamide (dFMRFa), and RYamide (RYa)−led to a significant reduction in egg-laying in *NP-GAL4 > UAS-TNT* flies compared to age-matched controls (*NP-GAL4 > UAS-TNTVIF*) (Fig 2A).

b. **No observable effects:** Interference with eight neuropeptides−allatostatin C (AstC), CCHamide-1 (CCHa-1), eclosion hormone (EH), hugin (hug-PK), myosuppressin (MS), pigment-dispersing factor (PDF), short neuropeptide F (sNPF), and tachykinin (TK)−did not significantly affect egg-laying under our assay conditions (Figs 2B and 2D).

**Table 3. Expression of neuropeptide GAL4s and egg-laying assay.** *NP-GAL4 > UAS-mCD8::GFP* expression in the brain and ovaries of adult female flies. Neuropeptide expression is denoted as '+' for presence, '-' for absence, and '?' for cases where data are either unavailable or require further validation. The egg-laying data shows the average number of eggs laid over a 24-hour period by controls (*NP-GAL4 > UAS-TNTVIF*) and experimental flies (*NP-GAL4 > UAS-TNT*), with the number of replicates noted in parentheses (*n*) and *p*-values for statistical comparisons. Egg-laying increase and decrease are represented as 'Δ' and '∇', respectively.

| | Neuropeptide | Stock | Expression | | Egg-laying | | |
|---|---|---|---|---|---|---|---|
| | | | Brain | Ovary | NP-GAL4> UAS-TNTVIF (n) | NP-GAL4> UAS-TNT (n) | p-values |
| 1 | AKH | #25683 #25684 | – – | – – | 63.85 ± 11.88 (27) 165.94 ± 14.53 (31) | 153.97 ± 10.06 (29) 130.71 ± 13.77 (31) | p < 0.0001Δ p = 0.084 |
| 2 | AstA | #51978 #51979 #80160 | + + + | + + ? | 306 ± 8.82 (16) 206.06 ± 8.24 (18) 186.47 ± 8.33 (15) | 237.43 ± 11.81 (16) 230.19 ± 4.37 (28) 171.19 ± 9.44 (16) | p < 0.0001∇ p = 0.016Δ p = 0.235 |
| 3 | AstB/MIP | #51983 | + | – | 264 ± 14.83 (13) | 35.92 ± 10.24 (13) | p < 0.0001∇ |
| | | #51984 | + | ? | 51984 > TNT - Larval lethality | | |
| 4 | AstC | #52017 | + | – | 285.83 ± 15.52 (24) | 251.32 ± 13.91 (27) | p = 0.094 |
| 5 | Burs | #51980 #40972 | – – | – + | 219.75 ± 9.95 (24) 279.71 ± 8.21 (24) | 189.33 ± 10.08 (24) 250.13 ± 11.83 (24) | p = 0.037∇ p = 0.046∇ |
| 6 | pBurs | #65470 | + | + | 65470 > TNT - Pupal lethality | | |
| 7 | CAPA | #51969 #51970 | – + | – – | 284.65 ± 8.28 (17) 251.94 ± 6.19 (16) | 189 ± 9.20 (17) 213 ± 8.97 (16) | p < 0.0001∇ p = 0.001∇ |
| 8 | CCHa-1* | #29266 | n/a | | 112.40 ± 14.43 (10) | 128.50 ± 14.62 (10) | p = 0.443 |
| 9 | CCAP | #25685 #25686 | + + | + + | 134.57 ± 12.31 (21) 117.05 ± 15.51 (20) | 85.62 ± 13.04 (21) 90.50 ± 12.34 (20) | p = 0.009∇ p = 0.189 |
| 10 | CRZ | #51976 #51977 | + + | – – | 241.94 ± 8.94 (16) 269.19 ± 6.09 (16) | 196.25 ± 13.95 (16) 196.81 ± 11.99 (16) | p = 0.011∇ p < 0.0001∇ |
| 11 | dFMRFa | #51990 #56837 | – – | – – | 263.63 ± 4.04 (16) 221.60 ± 19.01 (5) | 183.13 ± 11.38 (16) 53.80 ± 22.26 (5) | p < 0.0001 ∇ p < 0.0001∇ |
| 12 | DH31 | #51988 #51989 | + + | + + | 319.57 ± 16.26 (14) 252 ± 7.14 (22) | 272.40 ± 14.27 (15) 282.38 ± 8.07 (21) | p = 0.038∇ p = 0.007Δ |
| 13 | DH44 | #51987 | + | – | 305.17 ± 6.57 (18) | 222.17 ± 10.56 (18) | p < 0.0001∇ |
| 14 | DSK | #51981 | + | – | 263.92 ± 8.08 (24) | 220.96 ± 7.53 (24) | p = 0.0003∇ |
| 15 | ETH | #51982 | – | – | 51982 > TNT - Pupal lethality | | |
| 16 | EH | #6301 | + | – | 168.43 ± 10.94 (23) | 151.04 ± 15.06 (23) | p = 0.356 |
| 17 | hug-PK | #58769 | + | – | 255 ± 17.18 (26) | 244.56 ± 17.29 (18) | p = 0.670 |
| 18 | LK | #51992 #51993 | – + | – – | 238.19 ± 9.63 (16) 237.75 ± 5.38 (16) | 182.06 ± 17.09 (16) 241.38 ± 8.98 (16) | p = 0.008∇ p = 0.697 |
| 19 | MS | #51985 #51986 | – – | ? – | 266.08 ± 4.80 (25) 283.12 ± 10.55 (17) | 278 ± 9.17 (24) 261.9 ± 9.09 (20) | p = 0.257 p = 0.137 |
| 20 | PDF | #6899 | + | – | 142.59 ± 14.85 (22) | 175.23 ± 13.07 (22) | p = 0.106 |
| 21 | Proc | #51971 #51972 | + + | + + | 51971 > TNT and 51972 > TNT - Pupal lethality | | |
| 22 | RYa | #63899 | + | + | 237.94 ± 16.17 (16) | 130.88 ± 9.39 (16) | p < 0.0001∇ |
| 23 | sNPF | #51991 | + | – | 184.62 ± 11.74 (26) | 190.61 ± 9.90 (28) | p = 0.698 |
| 24 | SIFa* | #80696 #80697 #80698 | n/a | | 112.40 ± 14.43 (10) | 98.80 ± 11.62 (10) 144.80 ± 8.48 (10) 154.25 ± 7.70 (8) | p = 0.473 p = 0.073 p = 0.023Δ |
| 25 | TK | – | n/a | | 140.38 ± 18.81 (16) | 119.13 ± 14.70 (16) | p = 0.380 |

**Note:** *CS flies are used as controls for these neuropeptide mutant lines and n/a not applicable.

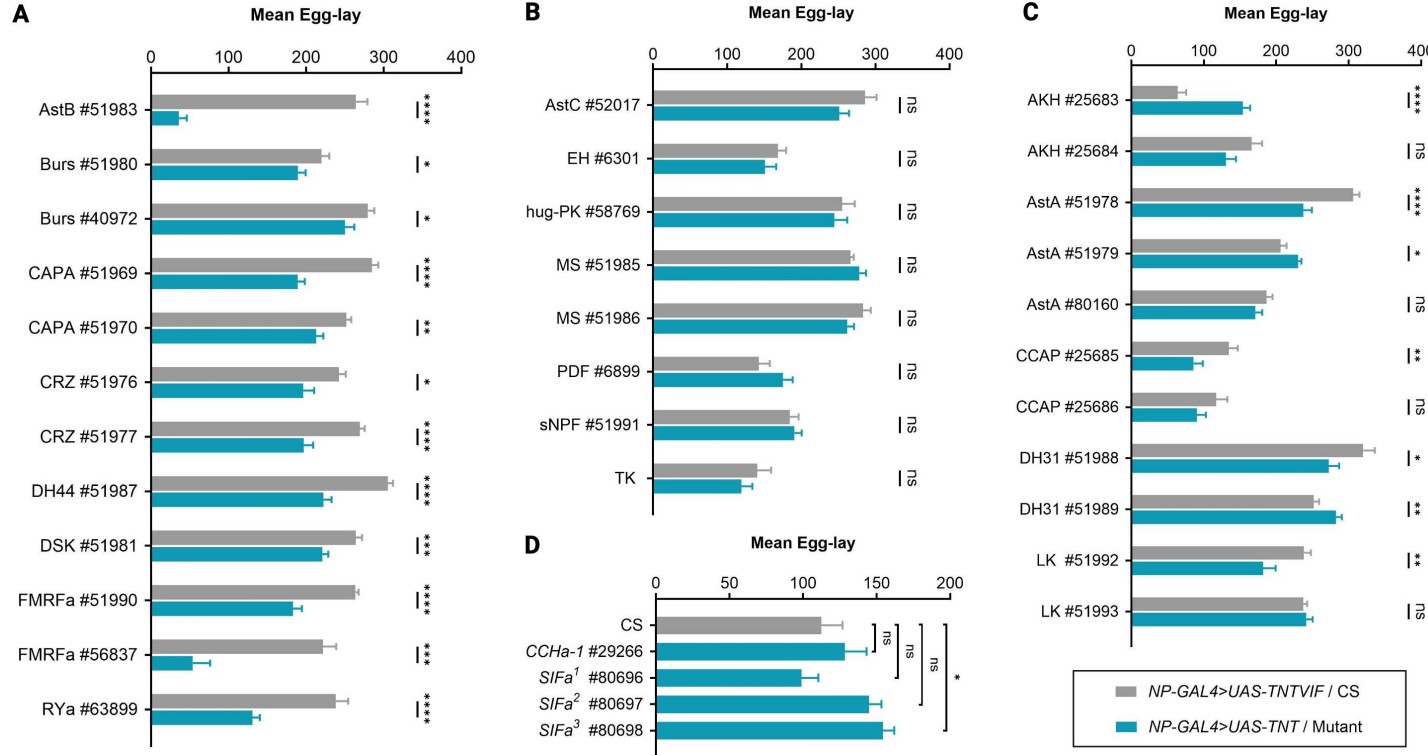

**Fig 2. GAL4-based screen for neuropeptides required in reproduction.** Histograms showing mean egg counts over a 24-hour period for **(A-C)** *NP-GAL4 > UAS-TNTVIF* and *NP-GAL4 > TNT* flies, classified into three categories: (A) reduced egg-laying, (B) no significant change, or (C) inconclusive results. *NP-GAL4* denotes the neuropeptide-specific GAL4 insertion line being tested. Refer to Table 1 for additional genotype information. **(D)** Egg-laying phenotypes for neuropeptide mutants–*CCHa-1* and *SIFa*. CS flies used as wildtype controls. Egg counts were measured as the number of eggs laid per five females. Data are presented as mean ± standard error of the mean (SEM). Statistical significance was determined using an unpaired *t*-test with Welch's correction. *p*-values are indicated as follows: $p < 0.05$ (*), $p < 0.001$ (**), $p < 0.0001$ (***), $p < 0.00001$ (****), and "ns" for non-significance ($p > 0.05$). See Table 3 for sample sizes (*n*) and *p*-values.

c.  **Inconclusive effects:** Six neuropeptides–adipokinetic hormone (AKH), allatostatin A (AstA), crustacean cardioactive peptide (CCAP), diuretic hormone 31 (DH31), leucokinin (LK), and SIFamide (SIFa)–produced inconsistent phenotypes across different GAL4 lines, precluding definitive functional conclusions (Figs 2C and 2D).

d.  **Developmental phenotypes:** Silencing of three neuropeptides–partner of bursicon (pBurs), ecdysis-triggering hormone (ETH), and proctolin (Proc)–caused developmental lethality, preventing the assessment of adult reproductive phenotypes (Table 3).

We tested one to three independent driver lines per neuropeptide to account for potential differences in insertion sites, expression patterns, and genetic background. Rather than selectively reporting only strong or consistent phenotypes, we document variability across GAL4 lines, highlighting the need for careful validation with complementary approaches and cautious interpretation in future mechanistic studies.

## Functional roles of neuropeptides in *Drosophila* reproduction

Beyond well-established reproductive regulators such as DILPs, SP, NPF, JH, and ecdysteroids, growing evidence demonstrates that numerous additional neuropeptides contribute to reproductive behaviors and physiology in *Drosophila*

[6,16,17]. For example, AstB/MIP regulates female post-mating responses [47,51]; CRZ controls male ejaculation [80,85]; DH31 promotes oocyte maturation through JH biosynthesis [21,102]; DH44 delays sperm ejection [109,116]; DSK modulates courtship behaviors [125,126]; and SIFa regulates mating dynamics [177,179]. Consistent with these studies, our screen corroborates roles of AstB/MIP, CRZ, DH31, DH44, DSK, and SIFa in female reproductive output [6,16] (Fig 2 and Table 3), validating the sensitivity of the assay.

**Allatostatin B (AstB)**, also known as myoinhibitory peptides (MIPs), are conserved ligands of the sex peptide receptor (SPR), which mediates female post-mating behavioral changes in response to male-derived SP transferred during copulation. The *mip* gene (CG6456) encodes five peptides (MIP1–5) expressed in the central nervous system (CNS) and intestine but absent from seminal fluid [10,192–195]. In our screen, *MIP-GAL4* drivers showed brain expression; however, neuronal silencing produced distinct outcomes. Driver #51983 significantly reduced egg-laying and showed ovary reporter expression, whereas driver #51984 caused larval lethality (Fig 2A and Table 3), likely reflecting differences in expression patterns and developmental requirements.

Functionally, MIPs relay SP signals from SPR-positive sensory neurons (SPSNs) innervating the uterine lumen to higher-order CNS circuits. Activation of female-specific MIP interneurons in the abdominal ganglion restores receptivity in mated females, whereas silencing these neurons reduces re-mating [51,53]. In addition, mating-induced SPR upregulation in polyamine-responsive chemosensory neurons–olfactory (IR41a and IR76b) and gustatory (IR76b)–further enables MIP signaling to shift food preference toward nutrient-rich diets that support reproduction [47]. SP–MIP signaling has also been implicated in mating-dependent long-term memory formation [52]. Collectively, these findings position AstB/MIP as a central neuromodulatory pathway integrating mating status, sensory input, and reproductive physiology.

**Corazonin (CRZ)** is a highly conserved neuropeptide related to AKH, with receptors homologous to the mammalian GnRH receptors [196–199]. In our screen, *CRZ-GAL4* expression was observed in the brain, and neuronal perturbation significantly reduced egg-laying (Fig 2A and Table 3). CRZ is essential for successful copulation in males; silencing four male-specific abdominal CRZ neurons prolongs copulation and disrupts sperm and seminal fluid transfer via serotonergic projection neurons innervating the accessory glands [80]. In addition, CRZ indirectly influences reproduction through energy mobilization and interactions with endocrine pathways involving JH, DILPs, NPF, and ecdysone signaling, all of which are central to reproductive behavior and physiology [200].

**Diuretic hormone 31 (DH31)** is a 31-amino acid calcitonin-like peptide involved in diuresis and in regulateing circadian temperature preference rhythms, locomotor activity, sleep, intestinal immunity, and the behavioral switch between feeding and courtship [93–102]. DH31 is expressed in brain neurosecretory cells and gut EECs. In our screen, both *DH31-GAL4* drivers showed reporter expression in the brain and low but detectable expression in mature ovarian follicles (Table 3 and S1 File). Neuronal silencing produced opposing effects on egg-laying: driver #51988 reduced egg-laying, whereas #51989 increased it (Fig 2C and Table 3). Similar discrepancies have been reported previously and attributed to impaired ovulation rather than defective oogenesis, leading to retention of mature oocytes [21].

Mechanistically, DH31-expressing brain neurons innervate the corpus allatum, where DH31-R activation suppresses JH biosynthesis. Reduced JH signaling induces reproductive dormancy by inhibiting oocyte maturation, directly linking DH31 signaling and reproductive state regulation [102]. DH31 also coordinates reproductive and feeding behaviors in response to nutrient availability: protein intake stimulates DH31 release from EECs, activating distinct DH31-R-expressing brain circuits that promote courtship via CRZ or suppress feeding via AstC [99]. Together, these findings suggest that DH31 regulate reproductive output by coupling nutritional state with ovulation and behavioral prioritization, rather than by directly controlling oogenesis.

**Diuretic hormone 44 (DH44)**, homologous to mammalian corticotropin-releasing hormone, is a 44-amino acid neuropeptide produced by neurosecretory cells in the adult brain [201,202]. In addition to its primary role in osmoregulation, DH44 modulates female reproductive physiology by regulating sperm storage. Following mating, females eject excess ejaculate several hours after copulation; DH44 signaling via DH44-R1 delays this process, promoting efficient sperm

storage and subsequent egg-laying [109]. Consistent with this function, silencing DH44-expressing neurons reduced egg-laying in our assay (Fig 2A and Table 3), likely due to impaired sperm retention and reduced fertilization efficiency [109,116]. DH44 also regulates sexually dimorphic and state-dependent behaviors through *doublesex* (*dsx*)-producing pC1 neurons, modulating locomotion, sexual arousal, and female sexual drive via CREB-dependent transcriptional mechanisms [107,115].

**Drosulfakinin (DSK)**, homologous to mammalian cholecystokinin (CCK), encodes two peptides (DSK-I and DSK-II) expressed in the brain, including a small subset of neurosecretory cells in the pars intercerebralis. DSK signals through two receptors–CCKLR-17D1 (CG42301) and CCKLR-17D3 (CG32540)–with distinct expression patterns in the brain and ventral nerve cord [118,203–206]. DSK neurons interact with sexually dimorphic *dsx* circuits to regulate mating behaviors in both sexes. In males, DSK-II suppresses sexual arousal via *fru^M*-expressing neurons [125], whereas in females, DSK signaling modulates receptivity through the pC1-DSK-MP1-CCKLR-17D3 circuit [126]. In our screen, silencing DSK-expressing neurons significantly reduced egg-laying (Fig 2A and Table 3). Given DSK's established role in mating behaviors and its co-expression in insulin-producing cells [117], this phenotype likely reflects indirect effects on reproductive output through altered mating efficiency or endocrine regulation.

**SIFamide (SIFa)** is produced by four large interneurons in the pars intercerebralis and exhibits extensive arborization throughout the adult CNS. Originally identified in flesh fly *Neobellieria bullata*, SIFa has conserved roles in feeding and reproduction and is functionally analogous to vertebrate gonadotropin-inhibitory hormone (GnIH) [180,207,208]. Using three previously validated CRISPR/Cas9-generated *SIFa* mutant lines [209], we observed increased egg-laying in one line, while the others showed no significant difference from wildtype controls (Fig 2D and Table 3). Although SIFa's reproductive functions remain incompletely defined, SIFa neurons integrate sensory and internal signals to modulate courtship circuits. RNAi-mediated knockdown of *SIFa* or SIFaR induces male-male courtship and increases female receptivity by acting on *fru*-expressing neurons [177,179]. Recent work further implicates SIFa-SIFaR signaling, together with CRZ pathways, in regulating context-dependent mating interval timing [210]. These findings suggest that SIFa influences reproductive output through modulation of mating dynamics rather than direct effects on oogenesis.

### *Drosophila* neuropeptides with potential role in reproduction

The findings above confirm known reproductive functions of several neuropeptides, validating the sensitivity of the assay. In addition, the screen identifies multiple neuropeptides with previously uncharacterized roles in reproduction.

**Adipokinetic Hormone (AKH)** is the insect functional homolog of mammalian glucagon, and plays a central role in regulating carbohydrates and lipid metabolism, thereby maintaining systemic energy homeostasis [211–213]. AKH signaling is well positioned to influence reproductive output by coordinating nutrient availability with egg production. In *Drosophila*, AkhR has been shown to regulate sex-specific reproductive behaviors in response to nutritional state, including male courtship activity and female sexual receptivity under starvation [214,215]. In our screen, silencing AKH-expressing cells produced variable egg-laying phenotypes: one driver line showed no significant change (#25683), whereas the other (#25684) resulted in increased egg-laying (Fig 2C and Table 3), suggesting that AKH may modulate reproduction indirectly by influencing energy allocation between somatic maintenance and reproductive investment. These findings are consistent with AKH acting as a neurometabolic integrator linking nutrient state to reproductive physiology.

**Allatostatin A (AstA)** peptides were originally identified as inhibitors of JH synthesis, a hormone essential for vitellogenesis and ovarian maturation [216]. In *Drosophila*, AstA neurons regulate feeding, growth, foraging, sleep, and insulin signaling, and project to both central and peripheral tissues, including insulin-producing cells. AstA signaling has been proposed to act upstream of reproductive maturation through its homology to the mammalian kisspeptin system, which governs puberty onset [45]. In our study, *AstA-GAL4* drivers showed strong expression in the adult brain and in two cases, innervation of the ovary (S1 File). However, silencing of AstA-expressing neurons resulted in variable egg-laying

phenotypes (Fig 2C and Table 3), suggesting that AstA may contributes to reproduction either by directly controlling or indirectly by integrating metabolic, endocrine, and developmental cues [216].

**Bursicon (Burs)** is a cystine knot neurohormone composed of two subunits encoded by *Burs* (CG13419, Burs α) and *pBurs* (CG15284, Burs β) [60–62]. It is primarily known for its role in post-eclosion cuticle tanning, and wing expansion [217–220]; however, emerging evidence from other insects implicates bursicon signaling in ovarian maturation and vitellogenesis [221,222]. In *Drosophila*, the bursicon receptor rickets is expressed in ovarian tissues and has been linked to boarder cell migration [223]. In our screen, silencing bursicon-expressing neurons reduced egg-laying without affecting adult viability (Fig 2A and Table 3), supporting a previously underappreciated role for bursicon in reproductive output, potentially through modulation of JH signaling or ovarian tissue remodeling.

**Capability (CAPA)** peptides are a family of diuretic neuropeptides encoded by the *Capa* gene, which produces three distinct neuropeptides: two periviscerokinins (CAPA-PVK1 and CAPA-PVK2) and one pyrokinin (Capa-PK). These peptides act through two GPCRs: *CapaR* (CG14575), which primarily responds to PVK1 and PVK2, and *PK1-R* (CG9918), a predicted receptor of CAPA-PK. CAPA-expressing neurosecretory cells are located in the abdominal ganglia and project to the corpora cardiaca and other visceral organs regulate stress response, fluid balance, and visceral physiology through neuroendocrine signaling [65–67]. While their direct role in *Drosophila* reproduction has not been defined, studies in other insects demonstrate that CAPA signaling influences egg production, hatching success, and survival rates [224]. In our screen, silencing of CAPA-expresssing cells significantly reduced in number of eggs laid despite the absence of ovarian expression, suggesting that CAPA peptides may act as indirect gonadotropic regulators, possibly by coordinating physiological state or stress responses with reproductive investment (Fig 2A, Table 3 and S1 File).

**Crustacean Cardioactive Peptide (CCAP)** is a conserved neuropeptide involved in ecdysis, cardiac regulation, and metabolic coordination. In *Drosophila*, CCAP neurons located in the brain and ventral nerve cord project to the reproductive tract [75–78] and are co-expressed with other neuropeptides implicated in reproduction, including Burs and MIP [92,225,226]. In our screen, both *CCAP-GAL4* lines showed reporter expression in the brain and ovaries (Table 3 and S1 File). However, silencing CCAP neurons reduced egg-laying in one driver line (#25685), indicating a potential reproductive role (Fig 2C and Table 3). CCAP may influence egg-laying indirectly through neuroendocrine signaling, regulation of muscle contractility in the reproductive tract, or coordination of metabolic state with reproductive timing.

***Drosophila* FMRFamide (dFMRFa)** peptides are widely expressed neuromodulators that regulate neuromuscular activity, hormone release, and behavioral states [87–92] (Table 2). Although not previously linked directly to oogenesis, dFMRFa-expressing neurons project to neurosecretory centers that control JH and ecdysteroid signaling, both of which are essential for egg production [9,227]. In our study, silencing dFMRFa neurons significantly reduced egg-laying (Fig 2A and Table 3), suggesting that these peptides may influence reproduction by modulating neuroendocrine pathways or reproductive tract physiology.

**Leucokinin (LK)** signaling integrates feeding behaviors, diuresis, stress responses, and locomotor activity [113,142–152]. Recent studies have also implicated LK in regulating female sexual receptivity and post-mating behaviors [228–230]. In our behavioral assay, silencing LK neurons produced inconsistent phenotypes across driver lines, likely reflecting context-dependent contributions of LK circuits (Fig 2C and Table 3). These results suggest that LK may influence reproductive output indirectly by coordinating internal physiological states and reproductive behavior rather than directly regulating oogenesis.

**RYamide (RYa)** is a recently identified neuropeptide with emerging roles in feeding suppression and water homeostasis [165,166]. Notably, in mosquitoes, RYa signaling coordinates feeding behavior with the gonadotrophic cycle by suppressing host-seeking following blood feeding and during egg development [231]. In *Drosophila*, we observed RYa expression specifically in mature oocytes (S1 File), and neuronal silencing significantly reduced egg-laying (Fig 2A and Table 3). These finding suggest that RYa may play a direct role in late-stage oocyte maturation or oviposition, identifying it as a strong candidate regulator of reproductive output.

### Neuropeptides without detectable reproductive phenotypes

**Allatostatin C (AstC)** regulates vitellogenesis and reproductive dormancy by linking mating status and environmental cues to JH synthesis [55,56,58]. Despite these established roles, silencing AstC neurons did not alter egg-laying in our assay, suggesting that AstC may regulate reproductive state transitions rather than short-term egg production (Fig 2B and Table 3).

**Pigment dispersing factor (PDF)** primarily functions in circadian regulation and male courtship behavior [158,232]. The absence of an egg-laying phenotype following PDF silencing is consistent with its limited role in female reproductive output (Fig 2B and Table 3).

**Ecdysis-triggering hormone (ETH)** is essential for development and adult reproductive physiology, including ovary maturation and egg production [127–132]. However, silencing ETH-expressing cells caused pupal lethality (Table 3), preventing assessment of adult reproductive phenotypes in this screen.

## Conclusion

This study presents a systematic functional screen of neuropeptides in *Drosophila* reproduction using neurogenetic perturbation and egg-laying as an integrated quantitative readout. Multiple neuropeptides significantly altered egg-laying, supporting a broad involvement of neuropeptidergic signaling in reproductive output. Rather than assigning definitive roles to individual neuropeptides, our findings emphasize the complexity and context dependence of these pathways, with observed phenotypes likely arising from combined effects on germline function, neuroendocrine signaling, mating behavior, and systemic physiology.

A central outcome of this work is the substantial variability observed across independent GAL4 insertion lines targeting the same neuropeptide, including inconsistent or opposing egg-laying phenotypes and mismatches between expression patterns and functional effects. By explicitly reporting these discrepancies, we highlight key challenges in reagent selection and interpretation, and raise important questions regarding indirect mechanisms, circuit-level contributions, and developmental versus adult functions. Similarly, reproductive phenotypes observed in the absence of detectable brain or ovarian expression suggest that many neuropeptides influence egg-laying indirectly through higher-order neural circuits, endocrine pathways, or physiological state, underscoring that egg-laying reflects an integrative reproductive output rather than oogenesis alone.

While egg-laying provides a robust and scalable measure of reproductive function, it inherently limits mechanistic resolution. In addition, genetic perturbations applied throughout development and assessment at a single time point constrain interpretation of temporal dynamics and compensatory effects. Overall, this study defines a functional landscape of neuropeptide involvement in *Drosophila* reproduction and, importantly, surfaces critical methodological and conceptual questions for the field. By documenting both consistent and inconsistent outcomes, it provides a transparent resource to guide reagent choice, experimental design, and targeted mechanistic follow-up.

## Supporting information

**S1 File. Neuropeptide expression in the ovary.** Representative whole-mount ovary images showing *NP-GAL4 > UAS-mCD8::GFP* expression, immunostained with anti-GFP (green), DAPI (blue), phalloidin (red). (PDF)

**S2 File. Raw egg-laying counts for all genotypes.** (XLSX)

## Acknowledgments

We thank Irene Miguel-Aliaga, Mani Ramaswami, and Bloomington *Drosophila* Stock Center for fly lines; Victoria L. Marlar and Diya Kashyap for assistance with fly pushing; and the Dartmouth Department of Biological Sciences Light Microscopy Facility for microscopy support. CM thanks Dartmouth's Academic Summer Undergraduate Research Experience

(ASURE) program. During preparation of this manuscript, the authors used ChatGTP to assist with minor language edits to enhance clarity and readability. All content was subsequently reviewed and revised by the authors, who take full responsibility for the accuracy and integrity of the publication.

## Author contributions

**Conceptualization:** Madhumala K. Sadanandappa.

**Formal analysis:** Madhumala K. Sadanandappa.

**Funding acquisition:** Madhumala K. Sadanandappa, Giovanni Bosco.

**Investigation:** Madhumala K. Sadanandappa, Caliope Marin, Shinae Park, Shivaprasad H. Sathyanarayana.

**Methodology:** Madhumala K. Sadanandappa.

**Project administration:** Madhumala K. Sadanandappa.

**Supervision:** Madhumala K. Sadanandappa, Giovanni Bosco.

**Visualization:** Madhumala K. Sadanandappa.

**Writing – original draft:** Madhumala K. Sadanandappa, Caliope Marin, Shinae Park.

**Writing – review & editing:** Madhumala K. Sadanandappa, Caliope Marin, Shivaprasad H. Sathyanarayana, Giovanni Bosco.

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
