## [Decision Letter · Decision Letter 0]

21 Dec 2025

PONE-D-25-58401GAL4-based functional screen of neuropeptides in Drosophila reproductionPLOS One

Dear Dr. Sadanandappa,

Thank you for submitting your manuscript to PLOS ONE. After careful consideration, we feel that it has merit but does not fully meet PLOS ONE’s publication criteria as it currently stands. Therefore, we invite you to submit a revised version of the manuscript that addresses the points raised during the review process.

We look forward to receiving your revised manuscript.

Kind regards,

Md Rajib Sharker, Ph.D.

Academic Editor

PLOS One

[This work was supported by the Human Frontier Science Program [LT000933/2017 to MKS] and the National Institute of Health [Pioneer grant 1DP1MH110234 to GB]].

Additional Editor Comments:

The manuscript is well-organized and clearly written, presenting a study objective that is both relevant and valuable for publication. However, I strongly encourage the authors to address several critical issues within the manuscript. These concerns need to be thoroughly addressed to enhance the quality and rigor of the study. Therefore, I recommend that the manuscript undergo significant revisions before it can be considered for publication.

Reviewers' comments:

Reviewer's Responses to Questions

**Comments to the Author**

1. Is the manuscript technically sound, and do the data support the conclusions?

Reviewer #1: Yes

Reviewer #2: Yes

2. Has the statistical analysis been performed appropriately and rigorously? 

Reviewer #1: Yes

Reviewer #2: Yes

3. Have the authors made all data underlying the findings in their manuscript fully available?

Reviewer #1: No

Reviewer #2: Yes

4. Is the manuscript presented in an intelligible fashion and written in standard English?

Reviewer #1: Yes

Reviewer #2: Yes

5. Review Comments to the Author

Reviewer #1: This study presents screening of 25 out of approximately 50 neuropeptides present in D. melanogaster for their role in egg-laying behavior. Authors employed a genetic technique that allowed them to interfere with NP signaling (expression of tetanus toxin in NP cells) to conclude about the involvement of NP in the egg-laying. Furthermore, transgenic flies carried a GFP tagged genetic constructs, which allowed for analysis of NP presence in the brain and ovaries. Results confirmed the role in reproduction for some NP and identified previously undescribed role for several NP in egg-laying.

This paper contains an impressive amount of work describing a gene expression of the NP constructs in the brain and ovaries and the behavioral phenotype. The experimental methods, including the number of samples and flies, show high standards. The outcome of each screened NP line on the phenotypes was described in terms of previous literature results, and this presents one of the major critiques of this paper. While partially useful it makes for a repetitive reading, without any major take-home messages. The other critiques is related to the previous one; authors have not described or tried to interpret few general observations/questions that arise from the work summary presented in the Table 2. These points are elaborated bellow.

1. It would help if authors described what was the rationale for picking these specific 25 NP, since they include some that have previously been described extensively, and some that have not. For example, the aim of the paper would have been clearer if they selected only those which have not previously described reproductive effect, or specifically, egg-laying defect, or another clear criteria.

2. Authors used NP-TNT flies of both sexes in the egg laying assay, which led to difficulty in interpretations of the phenotype – was it female or male derived effect, or a combination. Considering that expression analysis was done only in female brain and ovaries, but not testes, makes the interpretation even harder. Considering that egg laying assay is not overly time-demanding, it would be useful to repeat it with the control males (the same genetic background or inactive TNT). That way authors could with greater certainty ascribe phenotypes to the effect that NP have on female physiology.

3. How much does brain and ovaries staining contribute to the interpretations of egg-laying outcomes? Correlation between the presence in the ovaries and egg laying defect argue strongly for a direct role in egg-laying. But what about cases when there is no expression in the brain or ovaries, but the phenotype is present? My point is that the interpretations are hard and that one way of analyzing the data is to look for correlations and try to interpret them in the context of known literature. The interpretation should be in the context of commonalities that this screen showed, and not a review-like description of each single NP line, which makes it hard to see an overarching principle.

4. I compliment authors on pointing out the major limitations of this study in the Conclusion. Considering that phenotypes might reflect either direct or indirect role on egg-laying in females, or indirect effects on sperm production and behavior in males, it is hard to comprehend what is the major contribution of this paper. The fact that NP are important and that more mechanistic studies are needed is obvious. Maybe, the message could be clearer and simpler if the uncertainties of at least one factor are eliminated – males – by using wt/control males.

5. In the context of complexities and difficulties in explaining for the phenotypes I am not sure that the form in which the paper is written (L560/61 “experimental screening with a targeted literature review”) contributes to its clarity. Considering that based on this study a conclusive interpretation of the role that a specific NP has on egg-laying is impossible, the paper might be easier to read if it contained only the description of the results with important questions to be addressed. For example: inconsistencies or large differences on egg-laying between different insertion lines for the same NP, which raises the broader question about how to choose a best line to use (the one that leads to a phenotype that we like??), the inconsistencies between different NP (some show no expression in brain or ovaries but effect on egg-laying, some show the expression and the effect). So, expanding the paper with “literature review” does not help, but detracts from the main aim. Instead, authors should focus on the results and questions that this study raises.

Reviewer #2: In the article entitled ”GAL4-based functional screen of neuropeptides in Drosophila reproduction”, the authors investigated the potential role of 25 neuropeptides in reproduction. They assessed the number of eggs laid after disrupting the targeted neuropeptides and examined on parallel their expression in the brain and ovary.

The article is well-written, the experiment is well-designed and the conclusions are supported by the data. The two figures and three tables chosen to illustrate the data are relevant. However, I suggest modifying Figure 1, which lacks precisions for non-experts, and Table 3, which contains redundancies with Figure 2. In addition, the article could benefit from additional details to help non-experts better understand the strategy. Below are some comments to consider in order to improve the manuscript. I suggest reorganizing a little bit the discussion and shortening it by about half, since it is long and very descriptive, and most of the neuropeptide descriptions are actually already detailed in Table 2. Most of my remarks may be naive but I think that including some modifications would make the article accessible to a wider audience.

Refer to the attached file for more details

6. PLOS authors have the option to publish the peer review history of their article (what does this mean? ). If published, this will include your full peer review and any attached files.). If published, this will include your full peer review and any attached files.

**Do you want your identity to be public for this peer review?** For information about this choice, including consent withdrawal, please see our For information about this choice, including consent withdrawal, please see our Privacy Policy ..

Reviewer #1: No

Reviewer #2: No

---

## [Author Response · Author response to Decision Letter 1]

11 Mar 2026

Editorial Requests:

We have revised the manuscript to ensure compliance with PLOS ONE style requirements, including file naming conventions.

[This work was supported by the Human Frontier Science Program [LT000933/2017 to MKS] and the National Institute of Health [Pioneer grant 1DP1MH110234 to GB]].

We have included the statement, “The funders had no role in study design, data collection and analysis, decision to publish, or preparation of the manuscript" (lines 1272-1273) and have also included in the cover letter.

Not applicable.

4. Additional Editor Comments: The manuscript is well-organized and clearly written, presenting a study objective that is both relevant and valuable for publication. However, I strongly encourage the authors to address several critical issues within the manuscript. These concerns need to be thoroughly addressed to enhance the quality and rigor of the study. Therefore, I recommend that the manuscript undergo significant revisions before it can be considered for publication.

We thank the editor for the positive assessment of our manuscript’s organization, clarity, and relevance. We have carefully addressed all critical comments and clarified the study’s scope, reduced redundancy, emphasized key results and limitations, and improved overall clarity and rigor.

Reviewer #1: This study presents screening of 25 out of approximately 50 neuropeptides present in D. melanogaster for their role in egg-laying behavior. Authors employed a genetic technique that allowed them to interfere with NP signaling (expression of tetanus toxin in NP cells) to conclude about the involvement of NP in the egg-laying. Furthermore, transgenic flies carried a GFP tagged genetic constructs, which allowed for analysis of NP presence in the brain and ovaries. Results confirmed the role in reproduction for some NP and identified previously undescribed role for several NP in egg-laying.

This paper contains an impressive amount of work describing a gene expression of the NP constructs in the brain and ovaries and the behavioral phenotype. The experimental methods, including the number of samples and flies, show high standards. The outcome of each screened NP line on the phenotypes was described in terms of previous literature results, and this presents one of the major critiques of this paper. While partially useful it makes for a repetitive reading, without any major take-home messages. The other critiques is related to the previous one; authors have not described or tried to interpret few general observations/questions that arise from the work summary presented in the Table 2. These points are elaborated bellow.

We thank the reviewer for the thorough evaluation and constructive feedback. We have revised the manuscript to improve clarity, reduce repetition, and better articulate the rationale, scope, and key messages of this study.

1. It would help if authors described what was the rationale for picking these specific 25 NP, since they include some that have previously been described extensively, and some that have not. For example, the aim of the paper would have been clearer if they selected only those which have not previously described reproductive effect, or specifically, egg-laying defect, or another clear criteria.

This study was designed as an unbiased neurogenetic screen. We selected to categories of neuropeptides:

i. neuropeptides with known functions in reproduction (AstB/MIP, CRZ, DH31, DH44, MIP, DSK, SIFa, AstC, PDF, and ETH) to validate the robustness of our behavioral assay; with the exception AstC and PDF, these consistently reproduced known phenotypes, and

ii. neuropeptides for which reproductive roles had not been previously described, enabling discovery of novel association with reproduction.

We now clearly state this rationale, and the study aims in the revised manuscript (lines: 65-71; 245, 341-343)

2. Authors used NP-TNT flies of both sexes in the egg laying assay, which led to difficulty in interpretations of the phenotype – was it female or male derived effect, or a combination. Considering that expression analysis was done only in female brain and ovaries, but not testes, makes the interpretation even harder. Considering that egg laying assay is not overly time-demanding, it would be useful to repeat it with the control males (the same genetic background or inactive TNT). That way authors could with greater certainty ascribe phenotypes to the effect that NP have on female physiology.

We acknowledge that the observed phenotypes could arise from neuropeptide perturbation in females, males, or both. Expression analysis was performed only in female brains and ovaries, and we have clarified this limitation in the manuscript (lines: 200-207), describing these neuropeptides as having potential roles in reproduction that require further sex-specific studies.

All assays used males and females of the same genetic background; repeating the full screen with control males for all 25 genotypes is beyond the scope of this study. Nevertheless, our results provide initial insights and highlight candidates for future mechanistic investigation.

3. How much does brain and ovaries staining contribute to the interpretations of egg-laying outcomes? Correlation between the presence in the ovaries and egg laying defect argue strongly for a direct role in egg-laying. But what about cases when there is no expression in the brain or ovaries, but the phenotype is present? My point is that the interpretations are hard and that one way of analyzing the data is to look for correlations and try to interpret them in the context of known literature. The interpretation should be in the context of commonalities that this screen showed, and not a review-like description of each single NP line, which makes it hard to see an overarching principle.

The GFP expression analysis was performed primarily to validate the GAL4 reagents used in the screen. While correlations between expression in the ovary and egg-laying defects may suggest direct roles, the absence of expression in brain or ovary does not preclude indirect mechanisms. Without cell type-specific and functional dissection, definitive conclusion cannot be drawn. Therefore, we didn’t over stretch our findings and conclusions.

We have revised the text to emphasize that expression data a presented as descriptive context, not as direct functional evidence, and we avoid over-interpretation (Lines: 162-164). We also reduced review-like repetition and strengthened synthesis and shared patterns and limitations.

4. I compliment authors on pointing out the major limitations of this study in the Conclusion. Considering that phenotypes might reflect either direct or indirect role on egg-laying in females, or indirect effects on sperm production and behavior in males, it is hard to comprehend what is the major contribution of this paper. The fact that NP are important and that more mechanistic studies are needed is obvious. Maybe, the message could be clearer and simpler if the uncertainties of at least one factor are eliminated – males – by using wt/control males.

We thank the reviewer for this suggestion. The major contribution of this work is the systematic, side-by-side functional screening of a large fraction of Drosophila neuropeptides using a consistent genetic and behavioral framework. Beyond confirming roles of known reproductive neuropeptides, this study identifies several previously uncharacterized candidates affecting egg-laying and explicitly defines the limitations and future directions required for mechanistic resolution. We have clarified this contribution and scope in the revised manuscript (Lines: 87-90, 452-479).

5. In the context of complexities and difficulties in explaining for the phenotypes I am not sure that the form in which the paper is written (L560/61 “experimental screening with a targeted literature review”) contributes to its clarity. Considering that based on this study a conclusive interpretation of the role that a specific NP has on egg-laying is impossible, the paper might be easier to read if it contained only the description of the results with important questions to be addressed. For example: inconsistencies or large differences on egg-laying between different insertion lines for the same NP, which raises the broader question about how to choose a best line to use (the one that leads to a phenotype that we like??), the inconsistencies between different NP (some show no expression in brain or ovaries but effect on egg-laying, some show the expression and the effect). So, expanding the paper with “literature review” does not help, but detracts from the main aim. Instead, authors should focus on the results and questions that this study raises.

We agree that extensive literature review can detract from clarity when definitive functional conclusions cannot be drawn. In response, we have revised the manuscript to reduce per-neuropeptide literature discussion and removed language framing the study as a “targeted literature review.” The revised text now focuses primarily on the experimental results, explicitly highlighting key observations, inconsistencies across GAL4 insertion lines, and mismatches between expression patterns and egg-laying phenotypes.

Rather than selectively emphasizing consistent or strong effects, we intentionally report variability across genetic reagents to raise important questions regarding line selection, indirect versus direct mechanisms, and circuit-level contributions to reproductive output. We believe that documenting these discrepancies provides practical guidance for the community and aligns the manuscript more clearly with its primary aim as a functional screening study that defines candidates, limitations, and priorities for future mechanistic work.

Reviewer #2: In the article entitled ”GAL4-based functional screen of neuropeptides in Drosophila reproduction”, the authors investigated the potential role of 25 neuropeptides in reproduction. They assessed the number of eggs laid after disrupting the targeted neuropeptides and examined on parallel their expression in the brain and ovary.

The article is well-written, the experiment is well-designed and the conclusions are supported by the data. The two figures and three tables chosen to illustrate the data are relevant. However, I suggest modifying Figure 1, which lacks precisions for non-experts, and Table 3, which contains redundancies with Figure 2. In addition, the article could benefit from additional details to help non-experts better understand the strategy. Below are some comments to consider in order to improve the manuscript. I suggest reorganizing a little bit the discussion and shortening it by about half, since it is long and very descriptive, and most of the neuropeptide descriptions are actually already detailed in Table 2. Most of my remarks may be naive but I think that including some modifications would make the article accessible to a wider audience.

We thank the reviewer for the constructive evaluation and positive assessment of the study. We have revised Figure 1 to improve clarity for non-expert readers and modified Table 3 to remove redundancies with Figure 2. We have also added clarifying details to better explain the screening strategy and substantially reorganized the Discussion to reduce redundancy with Table 2 and improve accessibility, while maintaining a focused synthesis of the reproductive functions of Drosophila neuropeptides.

Abstract

Line 33. I am not sure to understand the meaning of the term “readout”. Would the term “trait” be more appropriate?

Thank you for the thoughtful suggestion. We have retained the term “readout” and clarified its usage in the revised manuscript by specifying “functional readout” to emphasize that egg-laying was used as a quantitative assay of reproductive output in this screen. We felt that the term "trait" could imply a fixed or inherent characteristic, whereas egg-laying here represents an outcome measure influenced by neuropeptide disruption (Line: 28).

Line 34. Replace implicated by “described or known to have a role”

We have incorporated the reviewer’s suggestion in line 29.

Introduction

Line 82. Here, I think you could explain the strategy and refer to Figure 1, which could include additional details. It was unclear to me how the constructs were obtained, and what role each of them played in the study. Since I am not familiar with the field, I tried to find more information in the references listed in the text but that was time-consuming.

Could you improve Figure 1 taking some good ideas from the following scheme?

Thank you for this helpful suggestion and for highlighting the perspective of non-Drosophila readers. We have revised Figure 1 as suggested and explicitly referred to it in the text (Lines: 66-67) to improve clarity and accessibility.

• All transgenic fly lines used in this study were obtained from the Bloomington Drosophila Stock Center, and detailed information on genotype, stock identifier, and chromosomal insertion for each neuropeptide is provided in Table 1. For additional background on how these stocks are generated and curated, we have directed readers to Flybase, which publicly maintains comprehensive information on transgenic fly stocks (Lines 98-101).

• The functional screen was conducted using the GAL4/UAS transgenic system, a well-established and widely used genetic approach originally developed by Andrea Brand and Norbert Perrimon (1993). We have cited the original publication to guide readers unfamiliar with this methodology and to provide the appropriate methodological context.

In your experiment, is Gal4 associated with gene X (neuropeptide) and UAS either with GFP (expression) or tetanus toxin (disruption of targeted gene?). It is unclear to me how this toxin works.

I suggest referencing the figure 1 here in the introduction part.

That is correct. While we briefly mentioned this approach in the Introduction, we provide a detailed description of the screening strategy and its rational in the Result section (Lines 158-165; 172-179). In this study, GAL4 drivers associated with each of the 25 neuropeptide genes were crossed with UAS transgenic lines for two complementary purposes:

i. NP-GAL4 X UAS-mCD8::GFP to visualize expression patterns in the brain and ovaries.

ii. NP-GAL4 X UAS-TNT to silence neuronal activity.

The tetanus toxin light chain (TNT) enzymatically cleaves synaptobrevin, a synaptic vesicle protein, thereby blocking synaptic vesicle release and effectively silencing targeted neurons. For controls, we used the inactive form of tetanus toxin (UAS-TNTVIF) under the same NP-GAL4 to assess any potential effects of toxin expression itself. By assessing phenotypic changes in neuropeptide-expressing neurons silenced in this manner, we were able to determine their functional importance in reproductive processes. These GAL4/UAS-based functional screening are well-established and widely used in the field.

Material and methods

Line 103. Table 1. Reference to Table 1 is only mentioned in this section. However, you do not explain why sometimes you used 1, 2 or three genotypes to one neuropeptides. Please explain the relevance. Do you think it is worth discussing some of your results concerning some genotypes where the gal-transgene is inserted in the X chromosome?

In addition to the M

---

## [Editor Report · Decision Letter 1]

12 Mar 2026

GAL4-based functional screen of neuropeptides in Drosophila reproduction

PONE-D-25-58401R1

Dear Dr. Madhumala Sadanandappa

We’re pleased to inform you that your manuscript has been judged scientifically suitable for publication and will be formally accepted for publication once it meets all outstanding technical requirements.

Kind regards,

Md Rajib Sharker, Ph.D.

Academic Editor

PLOS One

Additional Editor Comments (optional):

Authors have properly completed revision, and address all concerned issues. Now this paper can be considered for publication
---

## [Editor Report · Acceptance letter]

PONE-D-25-58401R1

PLOS One

Dear Dr. Sadanandappa,

I'm pleased to inform you that your manuscript has been deemed suitable for publication in PLOS One. Congratulations! Your manuscript is now being handed over to our production team.

Kind regards,

on behalf of

Dr. Md Rajib Sharker

Academic Editor

PLOS One